# Teaching and Learning Clinical Reasoning in Nursing Education: A Student Training Course

**DOI:** 10.3390/healthcare12121219

**Published:** 2024-06-19

**Authors:** Paula Leal, Ana Poeira, Diana Arvelos Mendes, Nara Batalha, Hugo Franco, Lucília Nunes, Fernanda Marques, Ljubiša Pađen, Małgorzata Stefaniak, Ana Pérez-Perdomo, Lore Bangels, Kathleen Lemmens, Guida Amaral

**Affiliations:** 1ESEL Nursing School of Lisbon, 1600-190 Lisbon, Portugal; leal2@esel.pt; 2Instituto Politécnico de Setúbal, Escola Superior de Saúde, Campus do IPS—Estefanilha, 2910-470 Setúbal, Portugal; ana.poeira@ess.ips.pt (A.P.); diana.mendes@ess.ips.pt (D.A.M.); nara.batalha@ess.ips.pt (N.B.); hugo.franco@ess.ips.pt (H.F.); lucilia.nunes@ess.ips.pt (L.N.); fernanda.gomes@ess.ips.pt (F.M.); 3Comprehensive Health Research Centre [CHRC], 1150-082 Lisbon, Portugal; 4Unidade Local de Saúde da Arrábida, EPE—Hospital São Bernardo, 2910-446 Setúbal, Portugal; 5Faculty of Health Sciences, University of Ljubljana, 1000 Ljubljana, Slovenia; ljubisa.paden@zf.uni-lj.si; 6Division of Nursing, Midwifery and Social Work, The University of Manchester, Manchester M13 9PL, UK; 7Faculty of Health Sciences, Medical University of Warsaw, 02-091 Warsaw, Poland; malgorzata.stefaniak@wum.edu.pl; 8Hospital Clinic of Barcelona, Fundacio Clinic per a la Recerca Biomedica, 08036 Barcelona, Spain; anperez@clinic.cat; 9University Colleges Leuven-Limburg, 3590 Diepenbeek, Belgium; lore.bangels@ucll.be; 10HBO Verpleegkunde Genk, 3600 Genk, Belgium; kathleen.lemmens@verpleegopleiding-genk.be; 11Comprehensive Health Research Centre [CHRC], 7002-554 Évora, Portugal

**Keywords:** clinical reasoning, nursing education, nursing students, teaching

## Abstract

Clinical reasoning is an essential component of nursing. It has emerged as a concept that integrates the core competencies of quality and safety education for nurses. In cooperation with five European partners, Instituto Politécnico de Setúbal (IPS) realized the “Clinical Reasoning in Nursing and Midwifery Education and Practice” project as part of the Erasmus+ project. As a partner, our team designed a multiplier event—the student training course. The aim of this report is to describe the construction and development of this clinical reasoning training course for nursing students. We outline the pedagogical approach of an undergraduate training course on clinical reasoning in 2023, which we separated into four stages: (i) welcoming, (ii) knowledge exploration, (iii) pedagogical learning, and (iv) sharing experience. This paper presents the learning outcomes of the collaborative reflection on and integration of the clinical reasoning concept among nursing students. This educational experience fostered reflection and discussion within the teaching team of the nursing department regarding the concept, models, and teaching/learning methods for clinical reasoning, with the explicit inclusion of clinical reasoning content in the nursing curriculum. We highlight the importance of implementing long-term pedagogical strategies in nursing education.

## 1. Introduction

Clinical reasoning is still a developing concept in nursing and midwifery. Recent studies have highlighted an existing challenge of not only understanding clinical reasoning as a concept but also teaching, assessing, and using it in clinical practice [1,2,3,4]. Furthermore, there is a general lack of literature and research evidence on clinical reasoning in the European Union context—such as understanding the clinical reasoning proficiency of recently graduated students in the European Union (education is harmonized in EU member states by directives)—and on teaching strategies employed within the EU. Consequently, there is a pressing need to comprehend how clinical reasoning is interpreted among students and registered nurses/midwives in the EU, including exploring teaching models and best practices in clinical reasoning education within EU universities and colleges.

To address some of these challenges, a partnership of six education and clinical institutions coordinated by the Hogeschool University of Applied Sciences (University of Leuven-Limburg, Belgium) developed the Erasmus+ Project “Clinical Reasoning in Nursing and Midwifery Education and Clinical Practice”—ERA + CRNM (agreement: 2021-1-BE02-KA220-HED-00023194, 2022–2024). This project brought together five European partners (Atlas College Genk, Belgium; Instituto Politécnico de Setúbal—Escola Superior de Saúde—nursing department, Portugal; Faculty of Health Sciences—University of Ljubljana, Slovenia; Faculty of Health Sciences—Medical University of Warsaw, Poland; Hospital Clinic of Barcelona, Fundación HED, Spain; and Hospital Clinic of Barcelona, Fundacio Clinic per a la recerca biomedical, Spain). Its aim was to build interconnectivity, innovation, inclusion, and digital competence in clinical reasoning in nursing and midwifery with the goal of understanding and building the connection between the clinical reasoning process in nursing and midwifery education and practices arises from this international work.

Throughout the ERA + CRNM project, each partner was assigned to organize multiplier events, including a student training course (STC). This paper describes the experience of developing the STC from the perspective of the Portuguese project team. The development of the STC leveraged the nursing department’s decision to formalize the transversal inclusion of the clinical reasoning process in the nursing degree curriculum.

### 1.1. The Conceptualization and the Potential of Clinical Reasoning in Nursing and Midwifery Education

The term ‘clinical reasoning’ has been used in nursing literature since the 1960s. In the 1980s, it was associated with critical thinking and referred to cognitive processes used by nurses when reflecting on patients’ problems. Clinical reasoning has been described as a sequential and interrelated process leading to a final decision through a series of logical inferences [5,6].

A review of the literature on clinical reasoning in nursing education [7] generally found little evidence of this in the nursing curriculum. Clinical reasoning, critical thinking, and clinical judgment are often used synonymously in the nursing literature [7,8]. However, they are not similar; critical thinking is a way of thinking that supports clinical reasoning, while clinical judgment results from the process of clinical reasoning [9,10].

We understand clinical reasoning in nursing students as a “holistic and recursive cognitive process with a dynamic, consistent and flexible nature to promote nursing students’ awareness and accurate perception of patients’ situations while enabling them to choose a course of action among different available alternatives and to put what they have learned from this situation into perspective” [7] (p. 6).

Nowadays, clinical reasoning is recognized as necessary in healthcare to meet the challenges of patients’ complex needs and as a skill that facilitates clinical decision-making [11,12,13].

Clinical reasoning is essential to nursing competence as it promotes critical, creative, scientific, and formal reasoning to ‘think like a nurse’ [4,14,15,16]. In this sense, nursing and midwifery schools should invest in curricula that focus on developing clinical reasoning to prepare students for the nursing care process of acquiring clinical reasoning [11,12,17,18]. The acquisition and development of clinical reasoning skills in nursing/midwifery will raise awareness of clinical reasoning in educational institutions and healthcare organizations.

### 1.2. Pedagogic Considerations in the Teaching/Learning Process of Clinical Reasoning

Developing clinical reasoning skills involves adopting long-term strategies in nursing education, emphasizing teaching and learning methods essential for improving problem-solving skills and self-efficacy in learning [8,10]. Reflection-based methodologies can facilitate the integration of theory and practice and enhance students’ clinical reasoning, namely simulation, case-based, and problem-based learning, journaling, collaborative learning, thinking aloud, clinical experiences, and mobile-based learning [2,8,19,20,21,22,23,24,25,26]. 

The literature mentions several aspects that may influence clinical reasoning learning by nursing and midwifery students: subject-specific knowledge, learning experiences, critical thinking, cognitive perception, intuitive abilities, affect, sleepiness, stress, problem-solving skills, academic self-efficacy, and lack of confidence [7,8,20,27,28]. In the European context, eight barriers to teaching clinical reasoning have been identified: time, culture, motivation, clinical reasoning as a concept, teaching, assessment, infrastructure, and others [3].

The challenges faced by students in learning clinical reasoning have been described by several authors: poor representation of the problem, difficulties in generating hypotheses, lack of clues that can help in gathering information, problems in interpreting information due to premature closure, inadequate prioritization of the issues, incomplete integration of information to form an accurate picture of the clinical situation, difficulties in formulating an adequate management plan, and poor communication skills [29,30]. Students described five themes regarding approaches they found beneficial or unsuitable for learning clinical reasoning. Beneficial approaches include practice with undifferentiated patients and teachers willing to make the thinking process explicit [30]. Unsuitable approaches include a lack of independence and participation, a lack of communication and feedback, and confusion caused by different sources of information. The findings suggest that the best way to develop clinical reasoning is through an active process of co-construction involving students, health professionals, and educators.

The theoretical and clinical learning environments and the role of the different actors in these contexts are essential for improving educational practice in clinical reasoning. Within this academic relationship, using student-centered methods and learning strategies is challenging for teachers. Wyngaarden, Leech, and Coetzee emphasize that educators need the resources, opportunities, and skills to use these strategies [31]. Research has demonstrated that educational strategies are more effective in developing clinical reasoning skills when implemented using a clinical reasoning model [2]. It is suggested that teaching strategies should be developed based on the components of clinical reasoning and used in an integrated way across the curriculum.

After considering all the evidence and mastering some student-centered pedagogical methods, we designed and implemented the STC to train first-year nursing undergraduate students in the acquisition/development of clinical reasoning skills.

## 2. Materials and Methods

### 2.1. Study Design and Participant Recruitment 

This study was carried out based on the student training course (STC) that took place on 26 June 2023 from 9:00 to 16:30 and was attended by 48 students from the first year of the nursing program. 

Pedagogically, the STC was organized into four stages: welcoming (9:00 to 10:00), knowledge exploration (10:00 to 12:30), pedagogical learning (13:30 to 14:30), and sharing experiences (14:30 to 16:30).

When welcoming the students, we carried out an icebreaker activity to promote a relaxed atmosphere and create the conditions to encourage the students to take part in the following activities. 

### 2.2. Data Collection

In terms of knowledge exploration, in the first step, we prepare an initial assessment of knowledge in clinical reasoning through a four-question quiz (one true/false and three multiple choice questions) using a digital gamification tool as a teaching resource, achieving 45.0% correct answers. In the second step, we introduced the World Café methodology to encourage reflection on the concept and application of clinical reasoning. The World Café methodology allows a structured learning and knowledge process and can also be used for exploratory qualitative research [4,32]. 

The World Café was structured using the following sequence: (i) dividing the students into groups; (ii) brainstorming with the teacher, who performs the role of facilitator; (iii) producing the outcome of the discussion; and (iv) sharing reflections in a large group.

(i)The students were randomly divided into six groups of eight.(ii)Each group had a facilitator who encouraged reflection on clinical reasoning based on two questions asked at two different times by the rotational dynamic of this methodology. The questions were: (1) What do you understand by clinical reasoning? and (2) In what situations have you applied or developed clinical reasoning? The groups discussed and reflected on the two questions in two separate 20 min sessions.(iii)The students organized the answers to the two questions in a graphic structure resembling a mind map created on paper using the materials provided by the facilitators at each table.(iv)The students shared the responses to the questions orally from each small group to the large group, encouraging discussion and allowing the facilitators to summarize the students’ main ideas. Killam et al. reinforce that general feedback and discussion through a large group help the facilitator to comment constructively on student contributions [33].

For pedagogical learning, the facilitation team had previously prepared a clinical case and a layout of the clinical reasoning model to facilitate the students’ learning. At this moment in the course, the clinical reasoning model was presented to the students using an expository method. Time was given to analyze the clinical case according to the stages of the clinical reasoning model presented. The facilitation team then provided information tailored to the needs and questions of the student group. This pedagogical learning was completed by the final assessment of knowledge in clinical reasoning, with the same quiz of four questions and two new questions (two true/false and four multiple choice) using the same digital gamification educational tool, obtaining 74.1% correct answers. 

When sharing their experiences of the STC, we asked students to express their pedagogical experiences regarding acquiring knowledge and competence and the relevance and satisfaction of attending this course. For the assessment, we built a five-point Likert scale questionnaire with six questions to assess how the students positioned themselves in relation to each statement regarding their level of agreement in terms of their satisfaction and the effectiveness and relevance of the course.

### 2.3. Data Analysis

The course evaluation data were analyzed using descriptive statistics principles to give us a correct and complete understanding of the relevance of the STC to this group of students. A thematic analysis of the graphic structures was carried out, similar to mind maps, produced by the students in the second phase of the STC using the World Café methodology. The students responded to two questions: (1) What do you understand by clinical reasoning? and (2) In what nursing situations have you applied or developed clinical reasoning? Two pairs of researchers analyzed the data using thematic analysis, as described by Braun and Clarke [34].

## 3. Results

Table 1 shows the results of the questionnaire designed to evaluate the course.

Regarding the course evaluation, the data show that 62.8% of the students fully agreed and 37.2% agreed that the STC enabled them to acquire new knowledge to apply clinical reasoning in clinical learning moments. Regarding the contribution of the presented model to the development of clinical reasoning skills in nursing, 72.1% fully agreed and 27.9% agreed with the learning approach. As for the relevance of clinical reasoning in nursing, 69.8% of the students considered it highly relevant, and 30.2% relevant. Also, 97.7% of the students agreed that clinical reasoning is applicable to learning and nursing practice.

Concerning the effectiveness of the STC, 41.9% of the students considered it highly effective, and 51.2% considered it effective. Regarding satisfaction with the STC, 48.8% were extremely satisfied, and 46.5% were satisfied. The overall evaluation of the STC showed that 55.8% of the students considered the course to be very good, and 39.5% considered it good.

The students’ answers in the World Café—the strategy used in the second phase of the STC—were analyzed using thematic analysis. Concerning the question “What do you understand by clinical reasoning?” six themes were constructed (Figure 1): essential tools of the profession; critical thinking to relate variables; systematized and cyclical reflection for decision-making; holistic and individualized nursing process; evidence-based practice to support knowledge and corroborate practice; and safety of care.

When analyzing the question “In what nursing situations have you applied or developed clinical reasoning?” seven themes were constructed (Figure 2): pedagogical strategies/simulation; learning in clinical teaching; nursing process; research; communication and transition of care; teamwork; and decision-making.

## 4. Discussion

This paper describes the construction and development of an STC on clinical reasoning for nursing students.

The results of the students’ evaluation of the course show that the STC enabled them to acquire new knowledge to apply clinical reasoning and that the learning approach promoted the development of clinical reasoning competencies. The results also highlight the relevance of clinical reasoning to nursing learning and practice. Furthermore, the students considered the STC effective and were satisfied with their attendance.

It is necessary to combine different teaching methods to train skills, and the choice and suitability of the techniques to the objectives can translate into the effectiveness of the students’ skills development [35]. The literature suggests combining teaching methods as different teaching–learning strategies can optimize learning outcomes, motivation, self-efficacy, responsibility, teamwork, and critical thinking, which underpin the clinical reasoning fundamental to decision-making [35,36]. In this article, we found that by using different methods—gamification, World Café, expository, and clinical case discussion—we could effectively develop clinical reasoning skills that are fundamental to clinical practice and increase student motivation, reflected in the satisfaction level. Different educational methods, such as group discussions, problem-solving, or simulation, significantly impact the students’ clinical reasoning skills [35,37].

Using the World Café method in the second phase of the STC made it possible to share ideas, explore different perspectives, reflect on concepts and structure, and engage in structured learning [4,32]. The students’ perspectives on clinical reasoning align with the concepts of clinical reasoning described in the literature.

The students highlighted in their perspectives on clinical reasoning the importance of critical thinking and consequent reflection [30] for decision-making [11,12,13]. Critical thinking supports clinical reasoning [9,10], which is often used synonymously in the nursing literature with clinical judgment [7,8]. However, clinical judgment results from the process of clinical reasoning [9,10], while clinical reasoning is a consistent, dynamic, and flexible cognitive process that promotes awareness and accurate perception of patients’ situations. This process makes it possible to choose between alternatives and reflect on the process itself [7]. Decision-making, supported by essential professional tools and evidence-based practice, makes it possible to provide individualized and safe care. Nurses’ use of clinical reasoning for decision-making—resorting to the underlying use of the profession’s basic instruments—and the consequent influence on patient outcomes is evidenced in the literature [5,6]. Competence in decision-making, based on clinical reasoning, is recognized as fundamental to responding to the complex needs of patients [11,12,13].

When analyzing the students’ perspectives on the applicability of clinical reasoning, we highlight the importance of teaching strategies applied systematically throughout the nursing course [8,10,38], such as concept mapping, case study, and simulation, supported by the nursing process for research-based decision-making, which enables the development of clinical, communication, and teamwork skills [11,12,13,18,38]. The choice and combination of teaching methods enhance the development of fundamental clinical reasoning for decision-making, self-efficacy, and teamwork [35,36]. Clinical reasoning supports the development of nurses’ skills and promotes formal reasoning to ‘think like a nurse’ [4,14,15,16]. Considering the aspects mentioned in the literature that can influence the learning of clinical reasoning by nursing and midwifery students [7,8,20,27,28], learning strategies based on reflection that increase problem-solving skills and self-efficacy in learning must be adopted systematically and consistently throughout nurses’ training [8,10,35].

### Limitations

To become a midwife in Portugal, you must complete a nursing course (1st cycle of higher education). Only after two years of professional practice as a nurse can one specialize in maternal health nursing and obstetrics. This midwife training is postgraduate, or 2nd cycle higher education, and seeks to provide professionals with specialized skills in healthcare for women, families, and the community in sexual and reproductive health. For this reason, in this class, where the present pedagogical methodology was applied, there were no postgraduate training students in midwifery, which is the major limitation of the present study.

In the future, it will be relevant to replicate the methodology in different years of the nursing degree course and compare possible differences in the conceptualization of clinical reasoning.

## 5. Conclusions

Clinical reasoning is a cognitive process that underlies clinical judgment and critical thinking and is essential for clinical decision-making. Therefore, acquiring metacognitive awareness by learning about clinical reasoning is fundamental for nursing students to be prepared to mobilize it in acquiring professional competence as future nurses. 

The results of the knowledge-assessment quizzes show a 29.1% increase in students’ learning about clinical reasoning (from 45.0% to 74.1%). In addition, all the themes that emerged in response to their reflections on the questions proposed when groups of students explored their knowledge were included.

The in-depth STC on clinical reasoning allowed us to expand our understanding and decide that this theoretical content should be explicitly integrated into the nursing curriculum. This would enable students to learn clinical reasoning progressively, from the simplest to the most complex decision-making.

The reflection on clinical reasoning underlying the development of this European project has opened different paths and had various impacts. One is to improve the pedagogical quality associated with methodological strategies in the teaching–learning process on clinical reasoning, and the other one is to contribute to the safety and quality of nursing care in the context of clinical practice.

## Figures and Tables

**Figure 1 healthcare-12-01219-f001:**
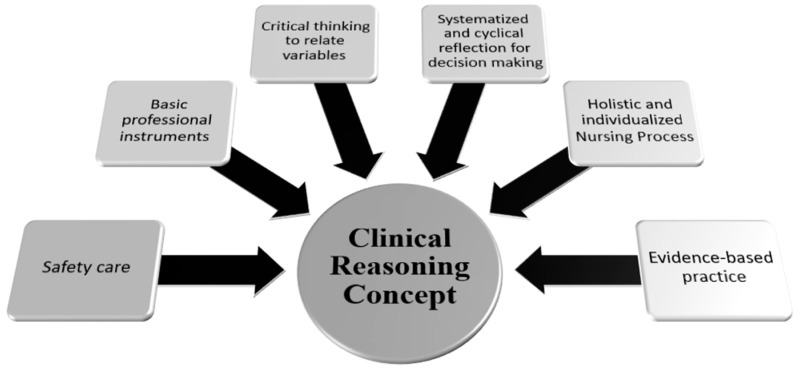
Clinical reasoning concept (source: authors).

**Figure 2 healthcare-12-01219-f002:**
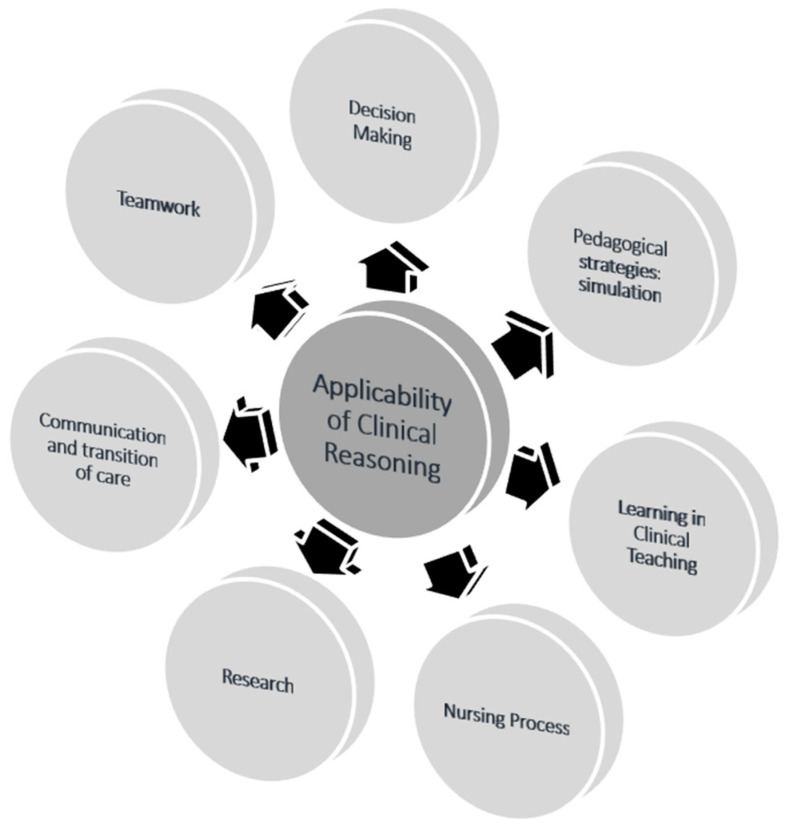
Applicability of clinical reasoning (source: authors).

**Table 1 healthcare-12-01219-t001:** Results of the evaluation of the STC.

Questions	Likert 5 Points Scale	Results n = 43 % (n)				
In your opinion, what is the relevance of clinical reasoning in nursing?	From 1 Not very relevant to 5 Highly relevant	1 0.0% (n = 0)	2 0.0% (n = 0)	3 0.0% (n = 0)	4 30.2% (n = 13)	5 69.8% (n = 30)
Do you agree that the presented model contributes to developing clinical reasoning competence in nursing?	From 1 Totally disagree to 5 Strongly agree	1 0.0% (n = 0)	2 0.0% (n = 0)	3 0.0% (n = 0)	4 27.9% (n = 12)	5 72.1% (n = 31)
What is your satisfaction with the STC in clinical reasoning?	From 1 Not at all satisfied to 5 Extremely satisfied	1 0.0% (n = 0)	2 0.0% (n = 0)	3 4.7% (n = 2)	4 46.5% (n = 20)	5 48.8% (n = 21)
In your opinion, what was the effectiveness of the training STC in clinical reasoning?	From 1 Not very effective to 5 Highly effective	1 0.0% (n = 0)	2 0.0% (n = 0)	3 7.0% (n = 3)	4 51.2% (n = 22)	5 41.9% (n = 18)
Do you agree that the STC in clinical reasoning brought you new knowledge that you can apply in your clinical learning?	From 1 Totally disagree to 5 Strongly agree	1 0.0% (n = 0)	2 0.0% (n = 0)	3 0.0% (n = 0)	4 37.2% (n = 16)	5 62.8% (n = 27)
What is your overall evaluation of the STC in clinical reasoning?	From 1 Very weak to 5 Very good	1 2.3% (n = 1)	2 0.0% (n = 0)	3 2.3% (n = 1)	4 39.5% (n = 17)	5 55.8% (n = 24)

## Data Availability

The data supporting this study’s findings are available from the corresponding author upon reasonable request. They are stored in controlled-access storage at the Instituto Politécnico de Setúbal.

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
