# Peer review of "Teaching and Learning Clinical Reasoning in Nursing Education: A Student Training Course"

_healthcare, 2024, doi:10.3390/healthcare12121219_

Round 1

Reviewer 1 Report

Comments and Suggestions for Authors

Understand that this is categorized as a brief report, the article summarized the results from an STC that aimed to develop clinical reasoning for nursing and midwifery students. Some suggestions are as follows:

Moderate editing work is needed throughout the whole article, also in particular:

The term student”s” training course is used in the title which is not the case in the main text.

The keywords nursing education / nursing students and midwifery / midwifery students are too similar, while pedagogical approach could be quite general.

Not sure if the content for the line 99 is not been able to shown.

Line 128-132 and line 134 – 136 are somehow repeated.

Wonder how 09:00-15:00 is 7 hours (line 147)

If “moments” is a specific term for the STC? (line 149)

Although this paper only describe part of a large project, it should clearly state what is included and what is not included in this paper. Is STC = world café? Section 2.2 discuss about the world café rather than the STC. Section 2.3 is then talking about data analysis of STC by descriptive statistics while later in the discussion mentioned about thematic analysis for the results of world cafe. This paper need to be readable and complete as a standalone work.

Not sure why descriptive statistics principles can “motivate students to better understand the importance of clinical reasoning” (line 199-200)

Section 2.4 – much of the content is not related to ethics.

Section 4 – need to talk about the implications from the questionnaire results.

Also, perhaps the authors can relate figure 1 to the literature review. Did the students conception for clinical reasoning mapping with the definition or the different dimensions of literature as described in line 72-98.

I do think the ending discussion is quite relevance (line 253-257). Perhaps this could supplement the ending sentence of abstract “The result was the explicit inclusion of clinical reasoning content in the nursing curriculum” which is a bit odd.

I just wonder some of the questions are taking too many cognitive load for first year nursing students. On one hand, only ~70% students get correct answers in the question set, on the other hand, they are asked “In what nursing situations have you applied or developed clinical reasoning?” Not sure they are really undergoing “nursing process” (figure 2).

Comments on the Quality of English Language

Moderate editing work is needed. 

Author Response

Dear Reviewer,

Thank you for your time, work, and contributions that will contribute to the publication of the high-quality paper.
Thus, we sent a table of discrimination with the changes made in the manuscript.
Also, all changes that resulted in an increase are highlighted in yellow in the manuscript. English correction changes are in review mode, using the Word tool.

Kind regards

Reviewer 2 Report

Comments and Suggestions for Authors

A great start but enhancement of the discussion section is certainly needed. I hope that my comments are useful to you. 

Comments on the Quality of English Language

quite good

Author Response

Dear Reviewer,

Thank you for your time, work, and contributions and references sent that will contribute to the publication of the high-quality paper.
Thus, we sent a table of discrimination with the changes made in the manuscript.
Also, all changes that resulted in an increase are highlighted in yellow in the manuscript. English correction changes are in review mode, using the Word tool.

Kind regards

Round 2

Reviewer 1 Report

Comments and Suggestions for Authors

Thanks for taking in much of my suggestions. I would like to clarify some of my concerns in the last round of review so that the authors can make their decision.

The term difference for "students training course" (line 3) and "student training course" are confusing (28, 68)

The new keyword "teaching method" is not specific as well.

It is mentioned in line 67 that (2) a world café, and (3) a student training course (STC) are separate events which confused the readers.

For the newly added limitations, it is mentioned that none of the student participants is undergoing midwifery education. It is hard to make a claim that this paper is a "training course for nursing/ midwifery students" (line 29). It is quite sure that students may not have the same increase in students' learning about clinical reasoning (from 45.0% to 74.1%) fir student who completed the nursing course and went through 2 years of professional practice. As a standalone publication, perhaps the authors have to delete all the description for midwifery instead.

Comments on the Quality of English Language

I strongly suggest the authors to seek help from professional editing services.

Author Response

Dear Reviewer,

We want to express our sincere gratitude for your time, work, and invaluable contributions. Your efforts will be reflected in the final version of the article and in the broader dissemination of knowledge.
Since the first revision round, we have made significant changes to the manuscript, all of which were guided by your valuable insights. We appreciate the opportunity you've given us to enhance the brief report about this methodology to develop clinical reasoning in nursing. 

Best regards.

Reviewer 2 Report

Comments and Suggestions for Authors

The participants were all nursing students; the authors still have not clarified for this reviewer the relationship between nursing students and midwifery ones. Why is midwifery addressed?

Comments on the Quality of English Language

some minor grammatical errors

Author Response

(The authors gave the same response as above.)
